# Estimated impact of RTS,S/AS01 malaria vaccine allocation strategies in sub-Saharan Africa: A modelling study

**Alexandra B. Hogan** *, **Peter Winskill**, **Azra C. Ghani**

MRC Centre for Global Infectious Disease Analysis, School of Public Health, Imperial College London, London, United Kingdom

* a.hogan@imperial.ac.uk

## Abstract

### Background

The RTS,S/AS01 vaccine against *Plasmodium falciparum* malaria infection completed phase III trials in 2014 and demonstrated efficacy against clinical malaria of approximately 36% over 4 years for a 4-dose schedule in children aged 5–17 months. Pilot vaccine implementation has recently begun in 3 African countries. If the pilots demonstrate both a positive health impact and resolve remaining safety concerns, wider roll-out could be recommended from 2021 onwards. Vaccine demand may, however, outstrip initial supply. We sought to identify where vaccine introduction should be prioritised to maximise public health impact under a range of supply constraints using mathematical modelling.

### Methods and findings

Using a mathematical model of *P. falciparum* malaria transmission and RTS,S vaccine impact, we estimated the clinical cases and deaths averted in children aged 0–5 years in sub-Saharan Africa under 2 scenarios for vaccine coverage (100% and realistic) and 2 scenarios for other interventions (current coverage and World Health Organization [WHO] Global Technical Strategy targets). We used a prioritisation algorithm to identify potential allocative efficiency gains from prioritising vaccine allocation among countries or administrative units to maximise cases or deaths averted. If malaria burden at introduction is similar to current levels—assuming realistic vaccine coverage and country-level prioritisation in areas with parasite prevalence >10%—we estimate that 4.3 million malaria cases (95% credible interval [CrI] 2.8–6.8 million) and 22,000 deaths (95% CrI 11,000–35,000) in children younger than 5 years could be averted annually at a dose constraint of 30 million. This decreases to 3.0 million cases (95% CrI 2.0–4.7 million) and 14,000 deaths (95% CrI 7,000–23,000) at a dose constraint of 20 million, and increases to 6.6 million cases (95% CrI 4.2–10.8 million) and 38,000 deaths (95% CrI 18,000–61,000) at a dose constraint of 60 million. At 100% vaccine coverage, these impact estimates increase to 5.2 million cases (95% CrI 3.5–8.2 million) and 27,000 deaths (95% CrI 14,000–43,000), 3.9 million cases (95% CrI 2.7–6.0 million) and 19,000 deaths (95% CrI 10,000–30,000), and 10.0 million cases (95% CrI 6.7–15.7 million) and 51,000 deaths (95% CrI 25,000–82,000), respectively. Under realistic

**Data Availability Statement:** The individual-based model of malaria transmission and the model parameter values are described in S1 Appendix. The transmission model code is available to

download at https://github.com/jamiegriffin/ Malaria_simulation. Malaria model outputs, code to run the ranking algorithm, and code to produce the figures and tables in the manuscript are available to download at https://github.com/mrc-ide/rtss_ prioritisation.

**Funding:** This work was funded by PATH URL: path.org/ (GAT.0888-07-06258-CRT) with additional support from the Bill and Melinda Gates Foundation (URL: http://www.gatesfoundation.org/ ) (OPP1068440). All authors acknowledge the MRC Centre for Global Infectious Disease Analysis (reference MR/R015600/1), jointly funded by the UK Medical Research Council (MRC) (URL: https:// mrc.ukri.org/) and the UK Department for International Development (DFID) (URL: https:// www.gov.uk/government/organisations/ department-for-international-development), under the MRC/DFID Concordat agreement and part of the EDCTP2 programme supported by the European Union. PATH contributed to the interpretation of results and drafting of the manuscript. All other funders of the study had no role in study design, data analysis, interpretation of findings, or drafting of the manuscript.

**Competing interests:** I have read the journal's policy and the authors of this manuscript have the following competing interests: PW discloses consultancy services to the Global Fund to support investment case and allocation modelling and country planning support. ACG discloses financial consultancy services to the Global Fund to support investment case and allocation modelling and country planning support and unrestricted research grants from a range of funders, including BMGF, UK Medical Research Council, The Wellcome Trust, NIH, Medicines for Malaria Venture, Integrated Vector Control Consortium, and Gavi. ACG is also a member of the WHO Malaria Policy Advisory Committee and of the Gavi Vaccine Investment Strategy Scientific Committee. ABH declares no competing interests.

**Abbreviations:** ACT, artemisinin-based combination therapy; admin-1, first administrative unit; CI, confidence interval; CrI, credible interval; DTP3, diphtheria, tetanus and pertussis vaccine dose 3; EIR, entomological inoculation rate; EPI, Expanded Programme on Immunization; IPTi, intermittent preventive treatment in infants; ISO, International Organization for Standardization; ITN, insecticide-treated net; LMIC, low- and middle-income country; MAP, Malaria Atlas Project; MVIP, Malaria Vaccine Implementation Programme; $PfPR_{2-10}$, *P. falciparum* prevalence in 2- to 10-year-olds; SMC, seasonal malaria

vaccine coverage, if the vaccine is prioritised sub-nationally, 5.3 million cases (95% CrI 3.5–8.2 million) and 24,000 deaths (95% CrI 12,000–38,000) could be averted at a dose constraint of 30 million. Furthermore, sub-national prioritisation would allow introduction in almost double the number of countries compared to national prioritisation (21 versus 11). If vaccine introduction is prioritised in the 3 pilot countries (Ghana, Kenya, and Malawi), health impact would be reduced, but this effect becomes less substantial (change of <5%) if 50 million or more doses are available. We did not account for within-country variation in vaccine coverage, and the optimisation was based on a single outcome measure, therefore this study should be used to understand overall trends rather than guide country-specific allocation.

## Conclusions

These results suggest that the impact of constraints in vaccine supply on the public health impact of the RTS,S malaria vaccine could be reduced by introducing the vaccine at the sub-national level and prioritising countries with the highest malaria incidence.

---

## Author summary

### Why was this study done?

- The RTS,S/AS01 malaria vaccine has previously been shown to be moderately efficacious in children, preventing approximately 36% of malaria cases in children who received 4 doses in a clinical trial.

- A pilot vaccine introduction is now ongoing in 3 African countries.

- Previous modelling has shown that implementing the vaccine could have a substantial public health impact and be cost-effective in preventing malaria cases and deaths in children.

- If the vaccine is recommended for wider introduction, it is likely that there will be an initial constraint on the number of doses available.

### What did the researchers do and find?

- We used an established model of malaria transmission to estimate the impact of the RTS,S malaria vaccine in sub-Saharan African countries, for different levels of RTS,S coverage.

- We applied a ranking algorithm to explore optimal vaccine allocation at the country and sub-national level, for different supply constraints.

- If initial malaria vaccine demand is higher than supply, prioritising the countries with the highest incidence would have the greatest impact in reducing malaria burden.

- Allocating the vaccine at the sub-national level was more efficient and allowed the vaccine to be introduced in almost double the number of countries compared to country-level introduction.

chemoprevention; WHO, World Health Organization.

## What do these findings mean?

- If the RTS,S vaccine is implemented beyond the pilot introduction, prioritising areas with the highest malaria burden may avert a substantial number of childhood deaths.

- Allocating the vaccine sub-nationally would maximise the overall public health benefit in terms of clinical malaria cases and deaths averted.

- Sub-national allocation would also allow more countries to introduce the vaccine earlier, ensuring more equitable access for populations at highest risk.

- Any sub-national introduction would involve challenges, and additional research would be needed to define locally appropriate metrics for vaccine prioritisation, including in the context of other malaria interventions.

## Introduction

Vaccines are one of the most successful and cost-effective interventions to reduce childhood mortality in low- and middle-income countries (LMICs) [1]. The establishment of Gavi, The Vaccine Alliance, in 2000 has helped to catalyse delivery of vaccines, supporting the World Health Organization's (WHO's) Expanded Programme on Immunization (EPI) programmes in 73 countries to vaccinate over 700 million children against major infectious diseases [2]. Across the 73 Gavi-supported countries, it is estimated that the vaccines delivered between 2001 and 2020 will avert more than 20 million deaths [3], whilst more recently it has been estimated that vaccines for 10 antigens implemented in 98 LMICs will avert 69 million (95% range 52–88 million) deaths between 2000 and 2030, the majority in children under 5 years of age [4]. The wider economic benefit of averting childhood deaths is also large—for a total estimated cost of US$20.8 billion over the period 2001–2020 [5], the wider economic return on this investment is estimated to be US$820 billion [3]. Investment in the development of vaccines for the remaining major childhood diseases therefore remains a priority.

Despite significant progress in reducing the burden of malaria over the past decade, progress has recently stalled [6]. Malaria is estimated to have caused 405,000 deaths in 2018, with 94% of these deaths occurring in sub-Saharan Africa and 67% in children younger than 5 years of age [7]. RTS,S/AS01 is the first vaccine targeting *Plasmodium falciparum* malaria that has demonstrated a protective effect in young children in a late-stage clinical trial. Phase III clinical trials for RTS,S were completed in 2014, with a 4-dose schedule administered at approximately 5, 6, 7, and 25 months demonstrating vaccine efficacy against clinical malaria of 36.3% (95% confidence interval [CI] 31.8–40.5) and efficacy against severe malaria of 32.2% (95% CI 13.7–46.9) in children aged 5–17 months at the first dose, over 48 months of follow-up [8]. However, due to concern regarding potential safety signals from the trial, lack of evidence of impact on deaths, and questions about the feasibility of delivery of 4 doses, in 2015, WHO recommended pilot implementation in order to resolve safety concerns and to establish sustained effectiveness, including impact on malaria hospitalisations and mortality [9]. The vaccine is now being evaluated in pilot introductions in Ghana, Kenya, and Malawi as part of the Malaria Vaccine Implementation Programme (MVIP), with approximately 360,000 children vaccinated each year over a 4-year period [10]. The pilot studies will obtain data on impact, safety, and feasibility (including an assessment of the incremental value of the fourth vaccine dose) [11], and these data will be used to inform a WHO policy recommendation about future use of

the vaccine. A positive policy recommendation would lead individual countries to make decisions about introducing the vaccine within existing malaria control programs and immunisation schedules. In terms of financing, vaccine roll-out in malaria-endemic regions may be considered for financial support by global agencies such as Gavi. RTS,S was included in the Gavi 2018 Vaccine Investment Strategy as a comparator vaccine and was assessed as providing value for money in comparison with other vaccines, although additional data—including on cost-effectiveness relative to other malaria interventions—would be needed to inform any future investment decision [12].

Many of the earliest vaccines introduced under the Gavi portfolio had been developed, tested, and delivered in high-income countries, and hence the evidence supporting vaccine introduction, including post-introduction safety monitoring, was well established. This greatly enhanced the speed with which these vaccines could be introduced and reduced the financial burden on the public healthcare sector. More recently, whilst some vaccines have become rapidly available (for example, haemophilus influenzae type b and hepatitis B vaccines), others (such as pneumococcal conjugate and human papillomavirus vaccines) have faced delays in implementation in LMICs compared to more rapid introduction in high-income settings [13]. Furthermore, whilst supply constraints often occur [14], for most vaccines some level of manufacturing capacity is available due to the existence of markets in high-income countries. Developing manufacturing capacity for a vaccine takes many years, and therefore a secure and funded market is required to ensure supply at introduction. Similar challenges are faced for vaccines for epidemic diseases in which stockpiling is required to ensure that supply is available when required. The challenges with retaining supply, as well as the public health consequences of insufficient supply, were clearly illustrated in the recent outbreak of yellow fever in Central Africa [15,16] and are common across other diseases (for example, cholera) [17]. While malaria is generally endemic in most locations (although epidemics do occur), uncertainty in future demand for a vaccine that has no high-income country commercial market means that it is likely that, at least initially, supply will be constrained [18].

In this study, we explore how limited vaccine supply could be targeted to ensure maximum health impact. Using a mathematical model of malaria transmission and the impact of interventions, we generate projections of vaccine impact in malaria-endemic regions in sub-Saharan Africa under different assumptions about the scale-up of other malaria interventions (remain at 2016 intervention coverage, or achieve WHO Global Technical Strategy targets) and for different vaccination scenarios (including coverage of 3 or 4 doses). Using these projections, we explore targeting of the vaccine to countries or to sub-national units to determine the optimal allocation under different constraints on vaccine supply in the first 5 years following vaccine introduction.

## Methods

### Mathematical model of malaria transmission and vaccine impact

We used a previously developed individual-based mathematical model of *P. falciparum* malaria transmission to estimate the impact of introducing the RTS,S malaria vaccine across Africa [19,20]. The model tracks the transmission of the parasite between humans and mosquito hosts (Fig A and Table A in S1 Appendix). In brief, individuals are born with a level of maternally acquired immunity which decays in the first 6 months of life, after which they become susceptible to infection from the bite of an infectious mosquito. Exposure depends on the entomological inoculation rate (EIR; the average number of infectious bites per person per unit time), which varies by location and is seasonally driven by rainfall patterns. On becoming infected, after a short latent period, individuals either develop clinical symptomatic disease

(the probability of which depends on their level of blood-stage immunity, which increases with age and exposure) or become asymptomatically infected. A proportion of those who develop clinical disease will develop severe pathologies—severe anaemia, cerebral malaria, respiratory distress, or other—at rates that depend on their prior exposure and age. These pathologies have an associated death rate. Those with clinical disease may seek treatment and—if successfully treated with first-line artemisinin-based combination therapies (ACTs)—enter a period of prophylaxis before returning to the susceptible state. Asymptomatically infected individuals recover from infection over a longer time period, with their detectability dependent on their level of immunity. Super-infection is incorporated, with asymptomatically infected individuals exposed at the same rate as susceptible individuals. All infected states are infectious to mosquitoes, with infectivity dependent on their level of detectability (as a surrogate for asexual parasite density). Mosquitoes become infected at a rate that depends on the infectivity of the human population and become infectious after a period of approximately 10 days, which reflects the extrinsic incubation period. The model has previously been parameterised by fitting to data on the relationship between EIR and parasite prevalence, clinical disease incidence, and severe disease incidence (Tables B and C in S1 Appendix). Full mathematical details and nongeographic parameter estimates are provided in S1 Appendix.

The RTS,S vaccine is modelled to reduce the probability of infection as it acts at the pre-erythrocytic stage of infection [21,22]. We modelled vaccine efficacy following the study by White and colleagues, with a bi-phasic pattern that simulates the initial rapid decay and a subsequent slower decay of vaccine-induced antibody titre, and a Hill function that captures the relationship between antibody titre and vaccine efficacy against infection over time. The efficacy function is given by

$$V(t) = V_{\max}\left(1 - \frac{1}{1 + \left(\frac{CSP(t)}{\beta}\right)^{\alpha}}\right)$$

where $V_{max}$ is the maximum efficacy against infection and $\alpha$ and $\beta$ are the fitted shape and scale parameters, respectively [23]. This model captures the dynamics observed in the phase II and III trials in children aged 5–17 months at the first vaccine dose [8,23]. Further details and parameter values are in S1 Appendix. We implemented the vaccine under a continuous EPI 4-dose schedule targeted at the 5- to 17-month age group, in which the first 3 doses are administered at months 6, 7.5, and 9 and the fourth dose at month 27, such that the first and third doses may align with routine child health appointments, and that the first dose at 6 months corresponds to the first scheduled RTS,S dose for the pilot program in Ghana and Kenya (noting that, in Malawi, the first RTS,S dose is scheduled at 5 months). Efficacy was assumed to occur following the third dose, in line with phase III trial data [8], and is boosted to a level similar to that achieved at the third dose for those that also receive a fourth dose 18 months post-dose 3.

Vaccine impact was parameterised using model fits to individual-level phase III trial data from 11 sites across Africa (Table D in S1 Appendix) [24]. This included follow-up for an average of 4 years following dose 3. We further validated our model by comparing site-specific predictions for vaccine impact in 3 of the phase III trial sites (Kombewa, Korogwe, and Nanoro) in which longer-term follow-up was monitored. Across the 3 sites, the model-predicted vaccine efficacy against clinical malaria in children aged 5–17 months was 21% over 7 years, closely matching the observed efficacy of 24% (95% CI 16–31) [25,26].

## Geographic sites and model outputs

We generated model outputs for all malaria-endemic African countries with at least one first administrative unit (admin-1) with *P. falciparum* prevalence in 2- to 10-year-old individuals ($Pf$PR$_{2-10}$) >10% based on Malaria Atlas Project (MAP) prevalence estimates for 2016 [27]. Twenty-eight countries comprising 464 admin-1 level units met this criterion. The individual-based model was calibrated to MAP prevalence estimates and clinical malaria cases obtained from the World Malaria Report 2017 [27,28]. Population growth was incorporated using United Nations projections [29], and country and admin-1 boundaries were sourced from GADM [30]. Historical interventions coverage estimates were derived from a range of sources: treatment with ACTs and indoor residual spraying implementation from the Demographic and Health Surveys, and insecticide-treated net (ITN) usage from MAP [27,31]. Map visualisations were produced using administrative boundary data from geoBoundaries [32].

## Intervention scenarios

We simulated 2 baseline intervention scenarios for future intervention coverage in the absence of vaccination (Table 1). The first scenario ("Maintain 2016 coverage") represents a continuation of current intervention coverage, whilst the second scenario ("High intervention coverage") represents increased coverage of interventions to levels similar to those modelled for the Global Technical Strategy for Malaria 2016–2030 [33,34]. These two baseline scenarios therefore provide reasonable bounds on the likely trajectories for malaria in the absence of vaccine introduction.

We then simulated 2 vaccination coverage scenarios, incorporating both the addition of vaccination and the coverage of other interventions in the baseline scenarios (giving 4 scenarios in total, Tables 1 and 2). In the first ("100% vaccine coverage"), every eligible child is assumed to receive the full vaccine schedule, whilst in the second ("Realistic vaccine coverage"), all eligible children are targeted, but vaccine take-up (and therefore effective coverage) is based on coverage of the third dose of the combined diphtheria, tetanus and pertussis vaccine (DTP3), using country-level DTP3 coverage from the WHO/UNICEF survey for 2017 [35]. Vaccination was assumed to be introduced in 2023, with 4 doses administered to 5- to 17-month-old children at 6, 7.5, 9, and 27 months of age. We did not account for buffer stock or wastage in vaccine distribution, due to the potential variability between countries.

For each scenario, we output the absolute number of clinical cases, severe cases, and deaths in children 0–5 years of age over time horizons of 5 and 10 years post-vaccine introduction. For the optimisation, posterior median parameter estimates from previous model fitting were

**Table 1. Baseline intervention scenarios.**

|  | **Maintain 2016 coverage** | **High intervention coverage** |
|---|---|---|
| **ITN coverage** | Continue ITN access from 2016 onwards | Increase ITN access to 80% by 2020, and 90% by 2025 and thereafter |
| **Indoor residual spraying** | Continue coverage from 2016 onwards | Continue coverage from 2016 onwards |
| **Treatment** | Continue coverage from 2016 onwards | Treatment with ACT increased to 80% by 2020 and thereafter |
| **SMC** | Continue coverage in recommended areas where SMC has been implemented from 2016 onwards [6] | Increase to 80% in recommended areas where SMC has been implemented by 2020 and thereafter [6] |

**Abbreviations:** ACT, artemisinin-based combination therapy; ITN, insecticide-treated net; SMC, seasonal malaria chemoprevention

**Table 2. Vaccine coverage scenarios.**

|  | 100% vaccine coverage | Realistic vaccine coverage |
|---|---|---|
| **Doses 1–3** | 100% | Coverage aligned with DTP3 with all doses still distributed [35] |
| **Dose 4** | 100% | 80% of coverage of doses 1–3 (values of 60% and 100% also simulated) |

**Abbreviation:** DTP3, diphtheria, tetanus and pertussis vaccine dose 3

used. Uncertainty in the impact was then incorporated by averaging outputs over 50 stochastic simulations with parameter draws from the posterior fitted distributions obtained from previous model fitting [20,36]. Clinical cases and deaths averted were calculated by comparing vaccine introduction scenarios to their respective baseline scenarios as the counterfactual.

## Optimising vaccine supply

Given a constraint in vaccine supply, we applied a steepest-descent algorithm to prioritise either countries or admin-1 units for vaccine delivery. For introduction at the subnational level, each admin-1 unit was treated independently of country. The algorithm was applied as follows: the number of events (clinical cases or deaths) averted per vaccine dose for each intervention scenario over a given time horizon was calculated and used to rank each country or admin-1 region in descending order. Beginning with the largest events averted per dose, countries or regions were then selected until the number of total vaccine doses required would exceed the vaccine supply level. Our primary optimisation outcome measure was the average clinical cases averted per year in 0- to 5-year-old children in the first 5 years following vaccine introduction. We also explored alternative outcome measures including a longer time horizon (average cases averted per year over 10 years) and an alternative outcome (deaths in 0- to 5-year-old children). In addition, we examined sensitivity to a lower vaccine coverage.

If all malaria-endemic countries decided to introduce the vaccine within areas with >10% $Pf$PR$_{2-10}$, we estimate the total vaccine dose demand to be of the order of 100 million doses per year for a 4-dose schedule in the first 5 years following vaccine introduction. We therefore performed the optimisation for a range of dose supply constraints between 10 and 60 million doses per year. We considered situations in which vaccine prioritisation decisions were made either at the country or admin-1 level. We also compared scenarios in which all 4 doses were administered (as per the current WHO recommendation) to scenarios in which only the first 3 doses were scheduled, or the country had the option of scheduling either 3 or 4 doses.

Under the MVIP, 3 countries (Ghana, Kenya, and Malawi) have recently begun pilot implementation of the vaccine. As these 3 countries would likely be prioritised for wide-scale rollout, we also considered a scenario in which vaccine supply was prioritised to these countries ahead of other malaria-endemic countries.

## Results

Under the "Maintain 2016 coverage" baseline intervention scenario, assuming 100% vaccine coverage of all 4 doses and that vaccine programmes are implemented at the country level, approximately 5.2 million (95% credible interval [CrI] 3.5–8.2 million) clinical childhood malaria cases and 27,000 deaths (95% CrI 14,000–43,000) could be averted per year based on a yearly dose constraint of 30 million (Fig 1, Table 3). If only 20 million doses are available, this is reduced to 3.9 million cases (95% CrI 2.7–6.0 million) and 19,000 deaths (95% CrI 10,000–30,000) averted, whereas 10.0 million cases (95% CrI 6.7–15.7 million) and 51,000 deaths (95% CrI 25,000–82,000) could be averted if 60 million doses are available, assuming in all scenarios

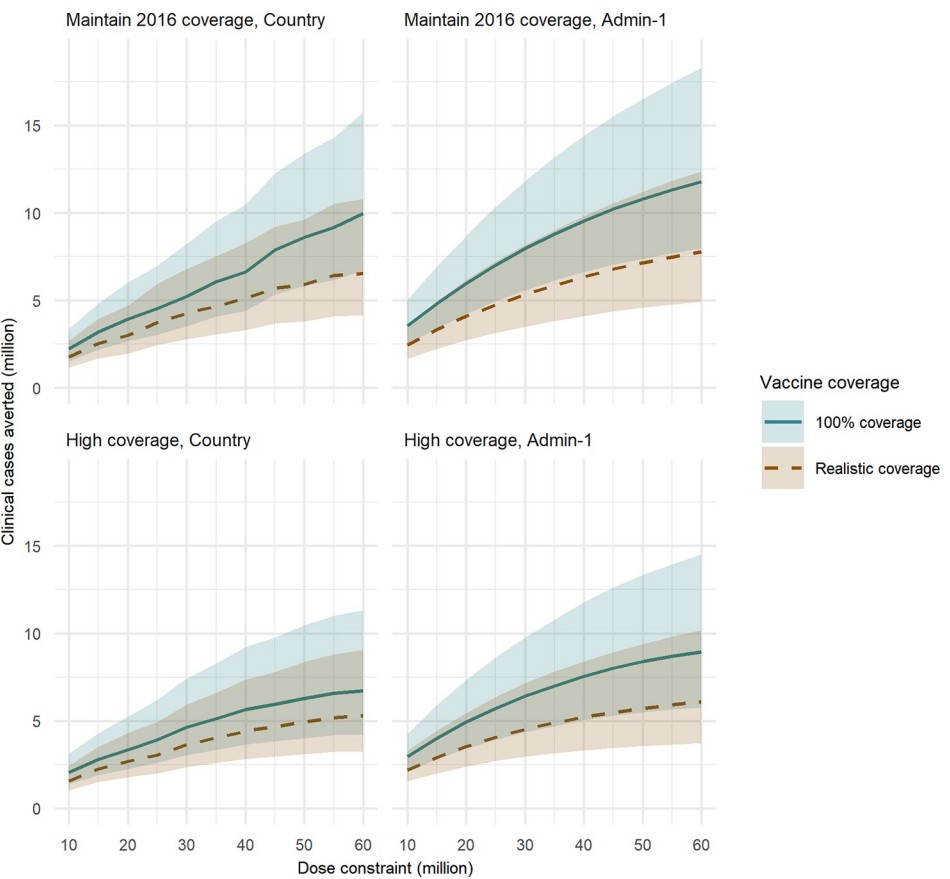

**Fig 1. Clinical cases averted for a range of vaccine dose constraints.** Total annual clinical cases averted in 0- to 5-year-old children in the first 5 years following vaccine introduction, for a range of annual dose constraints. (A) Optimised at the country level, "Maintain 2016 coverage" baseline intervention scenario. (B) Admin-1 level, "Maintain 2016 coverage" baseline intervention scenario. (C) Country level, "High coverage" baseline intervention scenario. (D) Admin-1 level, "High coverage" baseline intervention scenario. The "Realistic vaccine coverage" scenario is based on country-level DTP3 coverage for the first 3 vaccine doses, with coverage of the fourth dose set to 80% of that of dose 3. The shaded regions represent 95% CrI, based on 50 parameter draws. admin-1, first administrative unit; CrI, credible interval; DTP3, diphtheria, tetanus and pertussis vaccine dose 3.

that a vaccine program is rolled out at the country level (Fig 1, S2 Table). Under the same supply constraints, but under the "Realistic vaccine coverage" scenario (in which we continue to assume full coverage is distributed and the drop-off occurs in uptake), these impacts are reduced to 4.3 million cases (95% CrI 2.8–6.8 million)/22,000 deaths (95% CrI 11,000–35,000), 3.0 million cases (95% CrI 2.0–4.7 million)/14,000 deaths (95% CrI 7,000–23,000), and 6.6 million cases (95% CrI 4.2–10.8 million)/38,000 deaths (95% CrI 18,000–61,000) averted, respectively (Fig 1, Table 3 and S2 Table).

Assuming that the vaccine could be introduced at the subnational (admin-1) level rather than countrywide resulted in a greater estimated health impact, with 4.1 million clinical cases (95% CrI 2.7–6.2 million)/17,000 deaths (95% CrI 9,000–27,000) averted under a 20 million dose constraint, 5.3 million clinical cases (95% CrI 3.5–8.2 million)/24,000 deaths (95% CrI 12,000–38,000) averted under a 30 million dose constraint, and 7.8 million clinical cases (95% CrI 4.9–12.4 million)/41,000 deaths (95% CrI 19,000–64,000) averted under a 60 million dose constraint in the "Realistic vaccine coverage" scenario (Table 3 and S3 Table). These represent a 37%, 23%, and 18% increase, respectively, in clinical cases averted compared to the country-

**Table 3. Impact for 4 baseline intervention and vaccine coverage scenario combinations.** The impact is the annual events averted in 0- to 5-year-olds over 5 years following vaccine introduction, for a 4-dose schedule. Vaccine supply was constrained to 30 million doses per year. Additional dose constraints are in S2 Table and S3 Table. The 95% CrIs are based on 50 parameter draws. The countries introducing in each scenario are listed in alphabetical order. Three-letter codes for the countries are available in S1 Table.

| Baseline intervention scenario | Vaccine coverage scenario | Clinical cases averted in thousands (95% CrI) | Severe cases averted in thousands (95% CrI) | Deaths averted in thousands (95% CrI) | Clinical cases averted per 1,000 doses | Number of countries introducing | Countries introducing |
|---|---|---|---|---|---|---|---|
| *Country level* | | | | | | | |
| Maintain 2016 | Realistic | 4,254 (2,785–6,788) | 128 (63–205) | 22 (11–35) | 143 | 11 | BEN, BFA, COD, GAB, GHA, GIN, GNQ, MOZ, SLE, TGO, ZMB |
| Maintain 2016 | 100% | 5,234 (3,522–8,209) | 157 (78–245) | 27 (14–43) | 182 | 13 | BEN, BFA, CAF, COD, COG, GAB, GHA, GIN GNQ, LBR, MOZ, SLE, TGO |
| High | Realistic | 3,640 (2,354–5,924) | 113 (55–186) | 15 (7–24) | 122 | 11 | BEN, BFA, COD, GAB, GHA, GIN, GNQ, MOZ, SLE, TGO, ZMB |
| High | 100% | 4,639 (3,046–7,420) | 144 (71–229) | 19 (9–30) | 155 | 12 | BFA, CAF, COD, GAB, GHA, GIN, GNQ, LBR, MLI, MOZ, SLE, TGO |
| *Admin-1 level* | | | | | | | |
| Maintain 2016 | Realistic | 5,328 (3,502–8,162) | 143 (70–225) | 24 (12–38) | 178 | 21 | BEN, BFA, CAF, CIV, CMR, COD, COG, GHA, GIN, KEN, LBR, MLI, MOZ, MWI, NER, NGA, SLE, TCD, TGO, UGA, ZMB |
| Maintain 2016 | 100% | 7,958 (5,589–11,819) | 199 (102–310) | 34 (17–52) | 265 | 22 | BEN, BFA, CAF, CIV, CMR, COD, COG, GHA, GIN, GNQ, KEN, LBR, MLI, MOZ, MWI, NER, NGA, SLE, TCD, TGO, UGA, ZMB |
| High | Realistic | 4,512 (2,962–7,164) | 131 (65–208) | 17 (8–27) | 150 | 21 | BEN, BFA, BDI, CAF, CIV, CMR, COD, GHA, GIN, KEN, LBR, MLI, MOZ, MWI, NER, NGA, SLE, TCD, TGO, UGA, ZMB |
| High | 100% | 6,411 (4,348–9,779) | 178 (90–274) | 23 (12–55) | 214 | 20 | BEN, BFA, CAF, CIV, COD, GHA, GIN, GNQ, KEN, LBR, MLI, MOZ, MWI, NER, NGA, SLE, TCD, TGO, UGA, ZMB |

**Abbreviations:** admin-1, first administrative unit; BDI, Burundi; BEN, Benin; BFA, Burkina Faso; CAF, Central African Republic; CIV, Côte d'Ivoire; CMR, Cameroon; COD, Democratic Republic of the Congo; COG, Congo; CrI, credible interval; GAB, Gabon; GHA, Ghana; GIN, Guinea; GNQ, Equatorial Guinea; KEN, Kenya; LBR, Liberia; MLI, Mali; MOZ, Mozambique; MWI, Malawi; NER, Niger; NGA, Nigeria; SLE, Sierra Leone; TCD, Chad; TGO, Togo; UGA, Uganda; ZMB, Zambia

level introduction (Fig 1, Table 3, S2 Table and S3 Table). Subnational introductions also allowed the vaccine to be introduced in almost double the number of countries compared to national introduction (Table 3).

Fig 2 shows the prioritised countries for 6 levels of annual vaccine dose supply under the "Realistic vaccine coverage" scenario if the vaccine is introduced nationally. Under the most severe constraint (10 million doses), only 8 of the highest-incidence countries would introduce the vaccine (Benin, Burkina Faso, Congo, Equatorial Guinea, Gabon, Ghana, Sierra Leone, Togo). Notably, this does not include the 2 largest contributors to malaria burden (Nigeria and Democratic Republic of the Congo) because we assume that sufficient doses would be required to vaccinate children across all eligible areas (>10% $PfPR_{2-10}$) within each country before they are included. As the vaccine dose supply is increased, additional high-burden countries are generally included. For example, moving from 40 to 45 million doses in the "100% vaccine coverage" scenario leads to the inclusion of Nigeria (and results in the steep increase in clinical cases averted in Fig 1A). However, as vaccine dose supply increases, and if the vaccine is to be introduced at the national level in a new country, other countries may fall out of the optimal

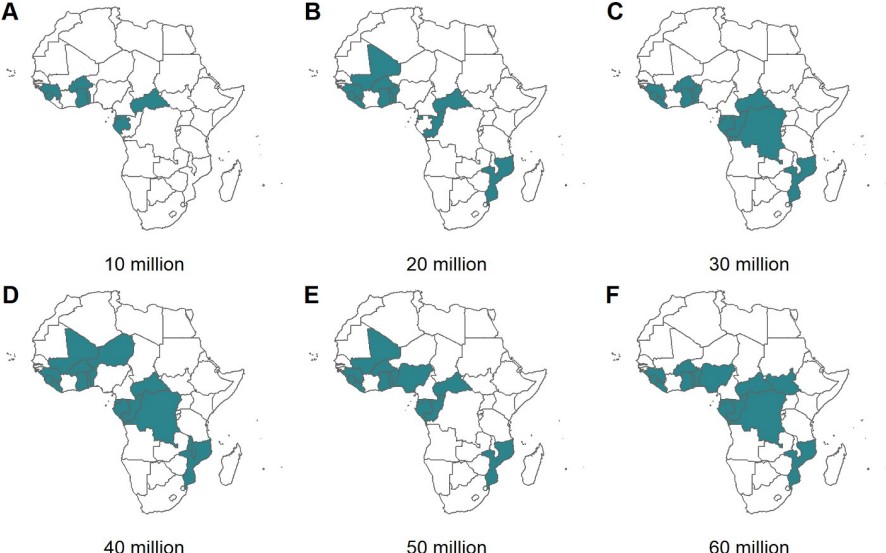

**Fig 2. Countries prioritised for vaccine delivery for a range of dose constraints, for the baseline intervention scenario of maintaining 2016 intervention coverage and realistic vaccine coverage.** The green shading represents prioritised countries for dose constraints of (A) 10, (B) 20, (C) 30, (D) 40, (E) 50, and (F) 60 million doses. Additional scenario combinations are in S1 Fig. The dose constraint is the maximum available RTS,S doses per year. The maps were prepared using administrative boundary data from geoBoundaries [32].

solution in order to cover the new large at-risk population. For example, assuming "Realistic vaccine coverage," moving from 20 to 30 million doses leads to the exclusion of Congo and Central African Republic in order to have sufficient doses to vaccinate the Democratic Republic of the Congo. Furthermore, the highest-burden country—Nigeria—is not included at 60 million doses. This is due to Nigeria currently having lower vaccine coverage rates compared to other countries which reduces the impact of introducing the RTS,S vaccine in this country. In contrast, if the vaccine is introduced sub-nationally, vaccine introduction occurs across a broader range of high-burden countries (Fig 3). The countries prioritised as supply increased were broadly similar under alternative assumptions about baseline intervention coverage (S1 Fig and S2 Fig).

If the 3 RTS,S pilot countries—Ghana, Kenya, and Malawi—are prioritised above all other countries, the overall events averted per vaccine dose is predicted to be lower compared to the fully optimal prioritisation (S4 Table and S5 Table). However, this difference in allocative efficiency reduces as levels of vaccine dose supply increase (Fig 4). In this scenario, under 2016 malaria levels and assuming 100% vaccine coverage, given a 30 million dose constraint and assuming that the vaccine is introduced sub-nationally elsewhere, we estimate that 6.9 million clinical cases (95% CrI 4.8–10.3 million) and 29,000 deaths (95% CrI 15,000–47,000) could be averted annually in 0- to 5-year-old children compared to 8.0 million clinical cases (95% CrI 5.6–11.8 million) and 34,000 deaths (95% CrI 17,000–52,000) for the same scenario in which the pilot countries are not prioritised (S5 Table, Table 3).

Under scenarios assuming 100% vaccine coverage, we found that implementing a 3-dose schedule was always suboptimal compared to a 4-dose schedule. Where vaccine take-up was reduced (under the realistic scenario), and particularly with lower levels of coverage of the fourth dose, the overall health impact was similar between the "either three or four doses" and the "four doses only" scenario, but the optimal dose strategy changed. Under the realistic vaccine coverage scenario, if the coverage of the fourth dose was either 60% or 80%, a 3-dose

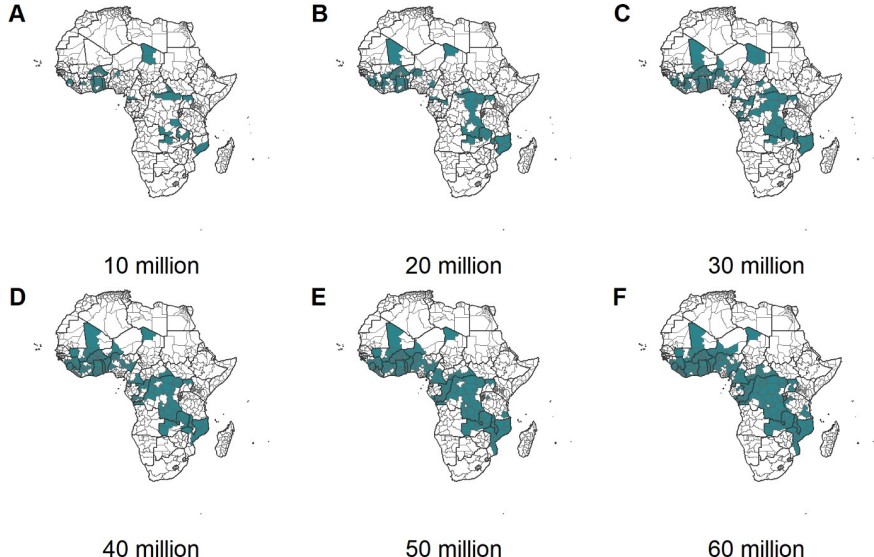

**Fig 3. Administrative units prioritised for vaccine delivery for a range of dose constraints, for the baseline intervention scenario of maintaining 2016 intervention coverage and realistic vaccine coverage.** The green shading represents prioritised admin-1 units for dose constraints of (A) 10, (B) 20, (C) 30, (D) 40, (E) 50, and (F) 60 million doses. Additional scenario combinations are in S2 Fig. The dose constraint is the maximum available RTS,S doses per year. The maps were prepared using administrative boundary data from geoBoundaries [32]. admin-1, first administrative unit.

schedule was often prioritised (Fig 5). However, in general, the same countries were prioritised. Allowing some countries to implement a 3-dose schedule also enabled the vaccine to be distributed across a larger number of countries (Fig 5). However, the difference between the total number of cases averted between the "four-dose" or "either schedule" options was small (S6 Table).

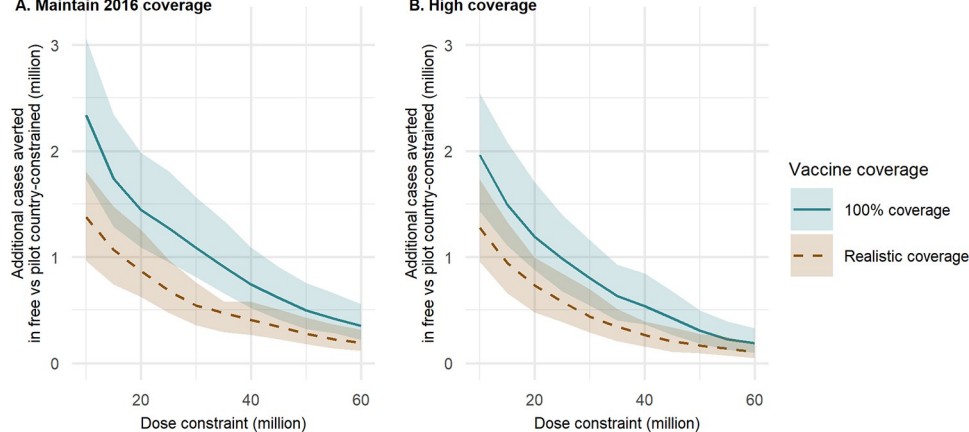

**Fig 4. The additional clinical cases averted when all doses are available compared to when the 3 pilot countries (Ghana, Kenya, and Malawi) are always prioritised.** Additional annual clinical cases averted in 0- to 5-year-old children in the first 5 years following vaccine introduction, for each of the baseline intervention scenarios: (A) "Maintain 2016 coverage" and (B) "High coverage." Dose constraints are optimised at the admin-1 level (outside of prioritisation countries). Two vaccine coverage scenarios are shown. The "Realistic coverage" scenario is based on country-level DTP3 coverage for the first 3 vaccine doses, with coverage of the fourth dose set to 80% of that of dose 3. The shaded regions represent 95% CrI, based on 50 parameter draws. Note that the total doses required at the lowest dose constraint (10 million) was 10.3 million in order to prioritise all 3 pilot countries. admin-1, first administrative unit; CrI, credible interval; DTP3, diphtheria, tetanus and pertussis vaccine dose 3.

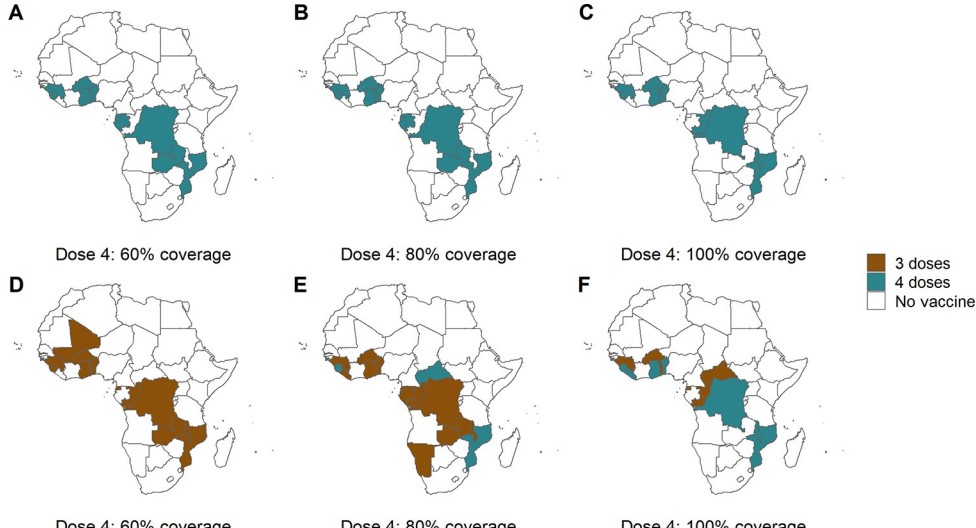

**Fig 5. Country-level vaccine allocation for different dose schedules and fourth dose coverage.** The upper row shows country allocation for a 4-dose schedule only, for 3 levels of coverage of the fourth dose as a proportion of third dose coverage: 60%, 80%, and 100% (A, B, and C). The lower row shows allocation where there is the option of either a 3- or 4-dose schedule, for the 3 levels of fourth dose coverage (D, E, and F). Coverage of the first 3 doses was based on DTP3 coverage in 2017, and the annual dose supply was limited to 30 million doses per year. The "Maintain 2016 coverage" baseline intervention scenario is shown, and additional results are in S6 Table. The maps were prepared using administrative boundary data from geoBoundaries [32]. DTP3, diphtheria, tetanus and pertussis vaccine dose 3.

We additionally considered different endpoints to prioritise vaccine delivery. This did not substantially change the list of prioritised countries for scenarios in which the vaccine dose constraint was reasonably high (above 30 million) and vaccine coverage was also high. However, when vaccine coverage is more severely constrained, small changes occur (S4 Fig and S5 Fig). Where deaths rather than clinical cases was considered as an endpoint, changes in country allocation were primarily determined by estimates of the access to care—such that the vaccine is prioritised for countries in which this is estimated to be low (if the aim is to prevent deaths) over countries with slightly higher malaria transmission but higher access to care (S4 Fig). Where a 10-year time period was considered, slightly fewer countries were allocated due to the impact of population growth (whereas vaccine dose supply was kept constant) (S5 Fig).

## Discussion

The RTS,S malaria vaccine has recently been introduced as part of a wide-scale implementation study in 3 countries in Africa—Ghana, Kenya, and Malawi. If remaining safety concerns are resolved and efficacy against severe disease is confirmed and a WHO recommendation is made, wider roll-out of the vaccine could commence from 2022 onwards [37]. During any initial expansion it is likely that, at least in the first instance, vaccine supply will be constrained as manufacturing capability is scaled up. Using an individual-based model of *P. falciparum* malaria and RTS,S vaccine impact, we estimated the clinical cases and deaths averted in sub-Saharan Africa following introduction of RTS,S, under several scenarios for vaccine coverage and the use of other interventions. We applied a prioritisation algorithm to identify vaccine dose allocation strategies—at both the country and subnational level—that would maximise the public health impact. Our results show that targeting of the vaccine to countries with the highest incidence—particularly those in the Sahel region—could have the greatest impact in reducing the burden of malaria.

The majority of childhood vaccines currently in the Gavi portfolio are implemented nationally. However, targeting of vaccine delivery does occur for some vaccines for which disease risk is geographically focal, such as during outbreaks of yellow fever and cholera. Furthermore, given the increased spatial heterogeneity of malaria burden, there is an existing precedent for targeting other interventions for malaria control, and a broader focus on data-driven subnational stratification of interventions within countries in order to make the most effective use of limited resources [38]. From a cost-effectiveness perspective, previous modelling has shown that the RTS,S vaccine would ideally be delivered to areas in which the parasite prevalence in children is greater than 10% [39]. However, the factors that would be considered to further prioritise introduction given a limited supply have not yet been identified. Our results demonstrate that greater public health impact can be achieved across a range of dose supply constraints if the vaccine is prioritised at the subnational level rather than nationally. Furthermore, targeting sub-nationally would enable a larger number of countries to partially introduce the vaccine early on, with expansion to other areas as supply constraints are eased, thereby making its introduction more equitable. This could be based on simple metrics such as burden or could incorporate other measures of equity such as the availability of alternative malaria interventions, prioritisation of those that do not have access to care, or other socioeconomic indicators. As the pilot introductions are being implemented at the subnational level, operationally such an approach should be feasible. However, any subnational introduction will entail additional challenges with planning, logistics, and public perception. Further research will be needed to establish locally appropriate metrics for prioritising subnational introduction and to ensure that this aligns with the stratification of other malaria interventions.

For our analyses, we assumed that once the vaccine is introduced either nationally or subnationally, the vaccine supply will be determined by the number of eligible children in that area. Under likely scenarios for coverage based on other childhood vaccines, this will inevitably mean that a proportion of doses will be wasted. One of the challenges identified with implementing the vaccine has been the need for a fourth dose given 18 months post dose 3, which may in some countries require a new vaccine contact point (although this could be combined with other health contact points). Our results suggest that, if coverage of this fourth dose is moderate, then in many countries it could be as efficient to introduce a 3-dose schedule as this would enable vaccine supply to a larger number of children. However, the public health benefits of doing so were estimated to be marginal (see S6 Table). Any future decision of this type would additionally need to consider the cost-effectiveness of the different schedules alongside the likely coverage levels that can be achieved. This should be informed by data from the pilot countries in the coming years.

Our vaccine model was calibrated to immunogenicity and efficacy data from the phase III clinical trial across 11 sites [23]. However, in the trial cohorts, usage of long-lasting ITNs was high, as was access to treatment [8], therefore we would anticipate vaccine effectiveness—as is being assessed in the pilot study—to reflect locally specific ITN uptake and treatment-seeking practices. We did not specifically model the impact of intermittent preventive treatment in infants (IPTi), which is recommended for delivery in some settings alongside routine immunisations, as IPTi implementation has been limited [7], but there is potential for this intervention to be more broadly introduced in the future. Therefore, estimates of RTS,S vaccine effectiveness in local settings with typical access to treatment and other malaria interventions will be important for planning broader vaccine introduction.

Prioritisation of countries based on deaths averted gave similar results to that based on cases averted, provided that the dose supply was not severely constrained. However, at the lowest dose levels, the countries selected differed between these 2 metrics. This is primarily because averting cases depends on the level of malaria transmission, whereas averting deaths depends both on the level of malaria transmission and the level of access to care. For a given

level of malaria transmission, in areas with better access to care, the vaccine is expected to have less of an impact on death because early access to treatment dramatically reduces the risk of severe malaria and thus death, as observed in the phase III trial in which there were very few deaths because of the high level of care provided to all participants. However, on average, access to care levels remains low across the continent [6,40], and therefore the absolute difference in impact is relatively low.

Our study has a number of limitations. First, the model is calibrated to match spatial estimates of malaria levels in the presence of other interventions, which are in turn modelled estimates based on varying data quality in different countries. Whilst these provide a good indication of variation between countries, they may not fully capture variation at finer spatial scales. Local data should also be used where available and may be particularly important to inform subnational introductions. Second, we did not model any vaccine-induced protection until after the third dose. It is possible that some protection would be conferred from the first 2 doses, but additional data would be needed to capture this impact in the model. Third, the optimisation is based on a relatively simple ranking algorithm that considers a single outcome (either cases or deaths averted) and does not incorporate other outcome measures such as cost-effectiveness. Fourth, our estimates of vaccine impact assume average coverage levels across the continent. This is clearly unrealistic as coverage will vary from one location to another within a country and may be correlated to access to other malaria interventions as well as access to health services more generally. The ongoing pilot implementation could be used to further inform these coverage assumptions. Fifth, for the 3 pilot countries, we did not incorporate the impact of the pilot study prior to 2023. In the pilot implementation, approximately 360,000 children will be vaccinated per year, requiring 1.44 million doses across their schedule, which represents 14% of our estimated eligible population in these 3 countries. However, accounting for these previously vaccinated children would likely reduce the projected impact in these 3 countries and therefore could bias the optimisation away from these countries. Finally, suboptimal scenarios can be indistinguishable from the optimal scenarios from a public health perspective because they are based on very small differences. Therefore, the outputs of this exercise should be considered in the context of understanding overall pattern rather than as directly guiding country-specific prioritisation.

With the plateau in malaria case estimates in recent years despite ongoing distribution of core vector control and chemoprevention interventions, the introduction of the RTS,S malaria vaccine has the potential to further reduce malaria cases and deaths in high-burden countries in Africa. If recommended by WHO for wider introduction from 2021 onwards, our results demonstrate that prioritising introduction in areas with the highest malaria burden has the potential to avert a significant number of childhood deaths and that subnational introduction could provide an equitable means to do so if operational issues can be overcome.

## Supporting information

**S1 Appendix. Mathematical model and parameters.** This content includes Fig A and Tables A to D.
(PDF)

**S1 Table. ISO country codes.** ISO, International Organization for Standardization.
(DOCX)

**S2 Table. Results summary of the prioritisation, for a range of dose constraints and intervention and vaccine coverage scenarios at the country level.** The impact is the annual events averted in 0- to 5-year-old children in the first 5 years following vaccine introduction, for the

4-dose schedule. 95% CrI represents the 95% credible interval, based on 50 parameter draws. The countries introducing in each scenario are listed in alphabetical order. Three-letter codes for the countries are available in S1 Table.
(DOCX)

**S3 Table. Results summary of the prioritisation for a range of dose constraints and intervention and vaccine coverage scenarios at the admin-1 level.** The impact is the annual events averted in 0- to 5-year-old children in the first 5 years following vaccine introduction, for the 4-dose schedule. 95% CrI represents the 95% credible interval, based on 50 parameter draws. The countries introducing in each scenario are listed in alphabetical order. Three-letter codes for the countries are available in S1 Table.
(DOCX)

**S4 Table. Results summary of the country-level prioritisation where the 3 pilot countries (Ghana, Kenya, and Malawi) were prioritised first.** The impact is the annual events averted in 0- to 5-year-old children in the first 5 years following vaccine introduction, for the 4-dose schedule. The relative impact is the clinical cases averted per 1,000 doses relative to that for the corresponding scenarios without pilot site prioritisation. 95% CrI represents the 95% credible interval, based on 50 parameter draws. Note that the total doses required at the lowest dose constraint (10 million) was 10.3 million, in order to prioritise all 3 pilot countries. The countries introducing in each scenario are listed in alphabetical order. Three-letter codes for the countries are available in S1 Table.
(DOCX)

**S5 Table. Results summary of the admin-1–level prioritisation where the 3 pilot countries (Ghana, Kenya, and Malawi) were prioritised first.** The impact is the annual events averted in 0- to 5-year-old children in the first 5 years following vaccine introduction, for the 4-dose schedule. The relative impact is the clinical cases averted per 1,000 doses relative to that for the corresponding scenarios without pilot site prioritisation. 95% CrI represents the 95% credible interval, based on 50 parameter draws. Note that the total doses required at the lowest dose constraint (10 million) was 10.3 million, in order to prioritise all 3 pilot countries. The countries introducing in each scenario are listed in alphabetical order. Three-letter codes for the countries are available in S1 Table.
(DOCX)

**S6 Table. Results summary for the analysis of different dose schedules and varying levels of fourth-dose coverage.** The impact is the annual events averted in 0- to 5-year-old children in the first 5 years following vaccine introduction. Ranking was performed at the country level for a dose constraint of 30 million doses available per year. Results are shown for the baseline intervention scenario "Maintain 2016 coverage." 95% CrI represents the 95% credible interval, based on 50 parameter draws. The countries introducing in each scenario are listed in alphabetical order. Three-letter codes for the countries are available in S1 Table.
(DOCX)

**S1 Fig. Countries prioritised for vaccine delivery assuming an annual dose constraint of 30 million doses and a 4-dose schedule.** Four combinations of the baseline interventions and vaccine coverage are shown. The colour gradient represents the average annual clinical cases averted in 0- to 5-year-old children per vaccine dose (CCA/dose) in the first 5 years post-vaccine introduction. Total annual events averted are reported in Table 3. The maps were prepared using administrative boundary data from geoBoundaries [32]. CCA, clinical cases averted.
(TIF)

**S2 Fig. Admin-1 units prioritised for vaccine delivery assuming an annual dose constraint of 30 million doses and a 4-dose schedule.** The colour gradient represents the average annual clinical cases averted in 0- to 5-year-old children per vaccine dose (CCA/dose) in the first 5 years post-vaccine introduction. Total annual events averted are reported in Table 3. The maps were prepared using administrative boundary data from geoBoundaries [32]. CCA, clinical cases averted.
(TIF)

**S3 Fig. Vaccine allocation where the 3 pilot countries, Ghana, Kenya, and Malawi, are prioritised.** Four combinations of the baseline interventions and vaccine coverage are shown. The allocation assumes an annual constraint of 30 million vaccine doses and is based on the 4-dose schedule, with allocation performed at the country level. The colour gradient represents the average annual clinical cases averted in 0- to 5-year-old children per vaccine dose (CCA/dose) in the first 5 years post-vaccine introduction. S3 Table shows the total events averted for each scenario combination and for additional dose constraints. The maps were prepared using administrative boundary data from geoBoundaries [32]. CCA, clinical cases averted.
(TIF)

**S4 Fig. Comparison of countries prioritised for vaccine delivery using two different ranking measures: Clinical cases averted per dose, and deaths averted per dose.** Results are shown for the "Maintain 2016 coverage" baseline intervention scenario and assuming "Realistic vaccine coverage," for a range of annual vaccine dose constraints: (A) 20 million, (B) 30 million, and (C) 40 million. The green shading represents countries prioritised for vaccine delivery.
(TIF)

**S5 Fig. Comparison of countries prioritised for vaccine delivery using two different ranking measures: Average annual clinical cases averted per dose over 5 years, and average annual clinical cases averted per dose over 10 years.** Results are shown for the "Maintain 2016 coverage" baseline intervention scenario and assuming "Realistic vaccine coverage," for a range of annual vaccine dose constraints: (A) 20 million, (B) 30 million, and (C) 40 million. The green shading represents countries prioritised for vaccine delivery.
(TIF)

**S6 Fig. Comparison of countries prioritised for vaccine delivery for three different levels of vaccine coverage.** The "100% coverage" and "Realistic coverage" are as described in Table 2, and the "Low coverage" scenario assumes that coverage of doses 1–3 is set at 75% of DTP3 coverage, with coverage of dose 4 set to 80% of dose 3. Results are shown for the "Maintain 2016 coverage" baseline intervention scenario (Table 1), for a range of annual vaccine dose constraints: (A) 20 million, (B) 30 million, and (C) 40 million. The green shading represents countries prioritised for vaccine delivery. Because even with suboptimal coverage we assumed that all vaccine doses are allocated, when we modelled a vaccine take-up that is still proportional to DTP3, the country allocation did not change compared to the "Realistic coverage" scenario. DTP3, diphtheria, tetanus and pertussis vaccine dose 3.
(TIF)

## Author Contributions

**Conceptualization:** Alexandra B. Hogan, Peter Winskill, Azra C. Ghani.

**Formal analysis:** Alexandra B. Hogan, Peter Winskill.

**Investigation:** Alexandra B. Hogan.

**Methodology:** Alexandra B. Hogan, Peter Winskill, Azra C. Ghani.

**Visualization:** Alexandra B. Hogan.

**Writing – original draft:** Alexandra B. Hogan.

**Writing – review & editing:** Alexandra B. Hogan, Peter Winskill, Azra C. Ghani.

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
