## [Editor Report · Decision Letter 0]

19 Mar 2020

Dear Dr Hogan, 

Thank you for submitting your manuscript entitled "Prioritising RTS,S/AS01 malaria vaccine allocation to maximise public health impact: a modelling study" for consideration by PLOS Medicine. Please accept my apologies for the delay in getting back to you about it. 

Your manuscript has now been evaluated by the PLOS Medicine editorial staff [as well as by an academic editor with relevant expertise] and I am writing to let you know that we would like to send your submission out for external peer review.

Kind regards,

Clare Stone, PhD,

PLOS Medicine

---

## [Decision Letter · Decision Letter 1]

6 May 2020

Dear Dr. Hogan,

Thank you very much for submitting your manuscript "Prioritising RTS,S/AS01 malaria vaccine allocation to maximise public health impact: a modelling study" (PMEDICINE-D-20-00728R1) for consideration at PLOS Medicine. 

[LINK]

In light of these reviews, I am afraid that we will not be able to accept the manuscript for publication in the journal in its current form, but we would like to consider a revised version that addresses the reviewers' and editors' comments. Obviously we cannot make any decision about publication until we have seen the revised manuscript and your response, and we plan to seek re-review by one or more of the reviewers. 

We expect to receive your revised manuscript by May 27 2020 11:59PM. Please email us (plosmedicine@plos.org) if you have any questions or concerns.

We look forward to receiving your revised manuscript. 

Sincerely,

Emma Veitch, PhD

PLOS Medicine

On behalf of Clare Stone, PhD, Acting Chief Editor,

PLOS Medicine

plosmedicine.org

*The abstract should be structured using the PLOS Medicine headings (Background, Methods and Findings, Conclusions -- Methods and Findings a single combined sub-section).

*In the Background section of the abstract, it would be good to also say something about the proven efficacy of the vaccine in trials, since the abstract just starts with the information about pilot implementation without saying anything about this being done against a background of the phase III trial data.

*In the last sentence of the Abstract Methods and Findings section, please describe the main limitation(s) of the study's methodology.

*At this stage, we ask that you include a short, non-technical Author Summary of your research to make findings accessible to a wide audience that includes both scientists and non-scientists. The Author Summary should immediately follow the Abstract in your revised manuscript. This text is subject to editorial change and should be distinct from the scientific abstract. Please see our author guidelines for more information: https://journals.plos.org/plosmedicine/s/revising-your-manuscript#loc-author-summary

*If possible, please reformat the in-text reference callouts so these are numbered citations in square brackets (eg [1, 2]) etc rather than superscript numerals. This should be quick and easy if referencing software was used. Many thanks. 

Comments from the reviewers:

Reviewer #1: The RTS,S/AS01 vaccine pilot is now well underway in three African endemic countries, and preparations need urgently to be made in view of possible large-scale implementation - in case the results of the pilot introduction are satisfactory. The present modelling study makes a good attempt at exploring some important parameters for future implementation, such as efficacy under different level of co-interventions, and at different levels of vaccine coverage. This study looks at two key public health parameters (clinical cases and deaths from malaria) and explores the very interesting issues of whether vaccine priority allocation should be considered at country or at sub-national level. The modelling seems to be solid and is done by one of the most experienced groups in the field.

Major comments

I have three major comments, which should be addressed before the manuscript can be considered for publication.

1. In the introduction on p5, the authors claim that the age groups for vaccination in the large-scale randomized controlled clinical trial done in 11 African sites and published in 2015 were "approximately 4, 5, 6 and 25 months". This puzzles me because the schedule in the RCT was 1,2,3, and 18 months. Since it was an RCT it was strictly enforced. The vaccination schedule at 5, 6 and 25 months is the one chosen for the pilot implementation in 3 countries in order to make it more EPI-compatible. In addition, because the vaccine is now implemented in a real-world situation, there will inevitably be some variability in the age at vaccination. 

Given how age-sensitive malaria incidence and mortality are, this difference must be considered. Since the vaccine parameters (especially age-efficacy) were derived from the RCT, the authors should mention this difference and have a clear explanation as to why it could matter or not. If it is likely to matter (which I believe), then an analytical approach should be designed to take that aspect into consideration. 

2. Modelling parameters for vaccine coverage were set to two levels: (1) current EPI levels for DPT3, and (2) 100% coverage. Informal evidence from the three pilot countries suggest that current uptake of RTS,S is much lower than for current EPI vaccines. There are a number of possible reasons for that. Without getting into a causal discussion, it seems that in addition to have a 100% coverage parameter it would be desirable to also have a scenario in which coverage is lower than the average EPI level for DPT3. 100% coverage is completely unrealistic, but that parameter value has the benefit of showing how good the vaccine could theoretically be. But to get a better sense of how valuable RTS,S would be if coverage was not as high as expected (which is now a very likely outcome of the current pilot implementation), modelling a lower uptake would be practically of great relevance. 

3. Discussion. No comment is made on the fact that distributing the vaccine over a maximum number of countries according to a sub-national impact optimization process is actually going to be very challenging. It is unlikely that countries will agree to such processes very easily. So while it is very good to show how this process of allocating at sub-national level can optimize limited resources, a caveat on the feasibility of this approach should be more clearly introduced in the discussion. The authors mention other GAVI-supported vaccines that are distributed according to risk areas. Giving such examples and showing possible analogies with the malaria vaccine situation would be very helpful and a good way of discussing this. 

Minor comments

References start with No 10, and lower number appear later. Presumably, this should be fixed?

Reviewer #2: See attachment

Michael Dewey

Reviewer #3: This paper addresses the important question of how to prioritise distribution of the RTS,S/AS01 malaria vaccine when there is limited supply. They use a well-established mathematical model to answer the question.

Specific comments:

1. Abstract: "we estimate 4.3 million malaria cases and 22,000 deaths in children U5 years could be averted per year at a dose constraint of 30 million. This decreases to 3.0 million cases / 86,000 deaths at a dose constraint of 20 million" Is this a mistake? Why does the number of deaths averted decrease with fewer vaccine doses?

2. Perhaps something went awry with the numbering of the references because the first one is number 10.

3. The authors consider two scenarios: "remain at 2016 intervention coverage, or achieve WHO Global Technical Strategy targets". Given the coronavirus pandemic, it seems very possible that coverage of malaria control interventions will decrease in the coming years. I wonder if the authors should consider a scenario where coverage of malaria control interventions decreases, leading to an increase in prevalence in 2-10 yearolds.

4. Methods paragraph 2: While I think it is reasonable to relegate the details of the mathematical model to the supplementary material, the authors should include the mathematical details of how vaccine efficacy is modelled in the main text.

5. The authors say "Efficacy was assumed to occur following the third dose". It is reasonable to assume that absolutely no protection is conferred by the first 2 doses?

6. Methods paragraph 6: I think "5-17-month-old children at 6, 7.5, 9 and 27 months of age" doesn't quite make sense. 27 months is not in the range of 5-17 months. What is the significance of 5-17-month here? Should it be 6-27 month old children?

8 Methods paragraph 8. I think the way countries or regions were chosen was quite simple. Could the authors try to explain it again, because I found the paragraph somewhat opaque. Using jargon such as "steepest-descent algorithm" does not, I think, aid clarity.

9 Results line 3: Should it be 157,000 rather than 157,00? If so, perhaps the authors could check the other numbers in the paragraph.

10 The number of 157,000 deaths averted (if that is correct) does not agree with the 27000 given in table 2. I think the numbers of deaths in the text correspond to the numbers of severe cases in the table.

11 Table 2: The description of the table does not say over what timescale the deaths/cases are averted. Does it assume 4 doses? 

12 At the end of the first paragraph of the results, table s1 is referenced. I think the authors mean table s2.

13 After multiple readings I still do not understand figure 5. Could the author please clarify? I don't understand it enough even to suggest an improvement, but one thing that is not clear to me is what is the difference between A, B and C (and likewise between D, E and F). 

14 Equation 21 in supplementary material. Is "max" in the wrong place?

[LINK]

---

## [Decision Letter · Decision Letter 2]

18 Aug 2020

Dear Dr. Hogan,

Thank you very much for re-submitting your manuscript "Prioritising RTS,S/AS01 malaria vaccine allocation to maximise public health impact: a modelling study" (PMEDICINE-D-20-00728R2) for review by PLOS Medicine.

I have discussed the paper with my colleagues and the academic editor and it was also seen again by two of the original reviewers. I am pleased to say that provided the remaining editorial and production issues are dealt with we are planning to accept the paper for publication in the journal.

[LINK]

We look forward to receiving the revised manuscript by Aug 25 2020 11:59PM. 

Sincerely,

Thomas McBride, PhD

Senior Editor 

PLOS Medicine

plosmedicine.org

AE comments:

The investigators used a previously developed individual-based mathematical model of P. falciparum malaria transmission, and a second model of RTS,S vaccine efficacy (White et al.), plus parameterization of the model also from White et al. 

My understanding is that the new mathematical modelling work and related code is the ranking algorithm where a range of vaccine coverage scenarios are explored, the rationale being that supply of the vaccine will be limited initially before manufacturing is scaled up.

A major driver for the current pilot studies in 3 countries was to look at the feasibility of implementing a 4th booster dose outside of EPI schedules which is predicted to be challenging. The investigators show the added protective efficacy conferred by this extra dose may not lead to a substantial impact on clinical malaria, if uptake is not high. The importance of this finding is not really stressed in the abstract or discussion- a 3 dose regimen has huge operational and cost advantages so I think it could be highlighted more.

Limitations of the analysis are that assumptions are not informed by any new efficacy or coverage data from the ongoing pilot studies and there is no cost effectiveness analysis 

Minor specific comments

1. The scenario where the WHO Global Technical Strategy targets are achieved references a modelling study by Griffin et al which considers SMC coverage but not IPTi, an intervention recommended by WHO which is also delivered with EPI (although uptake has been low to date). This could be mentioned as a limitation- although during the next 2 years of the COVID pandemic IPTi is unlikely to take off.

2. P11 in tracked changes version says: “The algorithm was applied as follows: the number of events (clinical cases or deaths) averted per vaccine dose.” Just checking it is per dose and not per course or per 3/4 initial doses since the model assumes no protection until 3 doses have been delivered. 

3. P12 “We also compared scenarios where all four doses were scheduled”. What does scheduled mean here? Received? Or means demand and dose not necessarily administered?

4. Typo “Diphtheria”

1- Please edit the Title to PLOS Medicine format: “Estimated impact of RTS,S/AS01 malaria vaccine allocation strategies in sub-Saharan Africa: a modelling study”

2- Please include the setting (in sub-Saharan Africa) in the early part of the Abstract Methods and Findings.

3- Abstract Methods and Findings, second sentence: “... in order to maximize events *averted*”?

4- Please include the 95% CrI ranges when quantifying outcomes in the Abstract.

5- You note that this vaccine has "moderate efficacy" (abstract), and is "efficacious in children" (author summary). In the introduction efficacy is quoted as 36.3%, which is pretty low (this is why the vaccine was not deployed 5 or more years ago). Please be more straightforward about the limited efficacy in the abstract/author summary, and perhaps mention the poorer efficacy in children (quoted as 24% in the methods section) in the Introduction.

6- Additionally, vaccine effectiveness in real life could be less, no? - this could be quoted as a limitation.

7- In the Abstract, you note "...sub-national prioritization (as currently planned)", could you describe the current prioritization plan for vaccine targeting in a bit more detail (the Introduction would be an appropriate place, and perhaps warranting an additional mention in the Discussion).

8- Please edit the Abstract Conclusions to limit the interpretation to the current study. “Our models estimate…” or “These results suggest…” could be helpful.

9- Perhaps you could comment somewhere (Introduction and/or Discussion) on any constraints on supplying and paying for RTS,S, given other vaccine programmes, existing and hoped-for.

10- The headings on Table 2 could be edited to make it clear the 95% CrI are in the parentheses but also relay tha numbers are X 1000. Similarly for the supplementary tables.

11- Could you include the credible intervals in the graphs in Fig 1 and 4?

12- In the first paragraph of the Discussion, please include a brief summary of what was done in this study before presenting your main findings. 

13- For reference 4 and any other preprints referenced, please include “Preprint” and the date cited.

“Li X, Mukandavire C, Cucunubá ZM, Abbas K, Clapham HE, Jit M, et al. Estimating the health impact of vaccination against 10 pathogens in 98 low and middle income countries from 2000 to 2030. Preprint. medRxiv. 2019. doi:10.1101/19004358v1 [cited Year, Month, Day]”

If the paper has since been published, please update the citation and check that the reference is still accurate.

14- Country codes are crowded on S1-3 Figs, consider removing.

Comments from Reviewers:

Reviewer #1: I am satisfied with the changes made by the authors in this second version of their manuscript. 

Reviewer #3: Thank you to the authors for addressing my comments.

[LINK]

---

## [Editor Report · Decision Letter 3]

25 Sep 2020

Dear Dr Hogan, 

On behalf of my colleagues and the academic editor, Dr. Elizabeth A Ashley, I am delighted to inform you that your manuscript entitled "Estimated impact of RTS,S/AS01 malaria vaccine allocation strategies in sub-Saharan Africa: a modelling study" (PMEDICINE-D-20-00728R3) has been accepted for publication in PLOS Medicine. 

PRODUCTION PROCESS

PRESS

PROFILE INFORMATION

Thank you again for submitting the manuscript to PLOS Medicine. We look forward to publishing it. 

Best wishes, 

Thomas McBride, PhD

Senior Editor 

PLOS Medicine

plosmedicine.org